# A Study of the Wear Mechanism of Composites Modified with Silicate Filler

**Sakhayana N. Danilova** [1] , **Sofia B. Yarusova** [2] , **Nadezhda N. Lazareva** [1] , **Igor Yu. Buravlev** [2] ,
**Oleg O. Shichalin** [3,*] , **Evgeniy K. Papynov** [3] , **Ivan G. Zhevtun** [2] , **Pavel S. Gordienko** [2]
**and Aitalina A. Okhlopkova** [1]

1   Chemical Department, Institute of Natural Sciences, North-Eastern Federal University, Yakutsk 677000, Russia
2   Institute of Chemistry, Far Eastern Branch, Russian Academy of Sciences, Vladivostok 690022, Russia
3   Department of Nuclear Technology, Institute of Science-Intensive Technologies and Advanced Materials,
    Far Eastern Federal University, Vladivostok 690922, Russia
*   Correspondence: oleg_shich@mail.ru

**Abstract:** The article considers the effect of a filler based on synthetic wollastonite ($CaSiO_3$), which is introduced into a polymer matrix made of ultra-high molecular weight polyethylene, on the tribotechnical parameters of the produced polymer composite material. Behavioral features of composites after friction were investigated by infrared spectroscopy and scanning electron microscopy. It was found that the introduction of wollastonite into the polymer matrix contributed to a reduction in the friction coefficient by 23% and the wear rate by four times. In the micrographs of the friction surfaces of the obtained composite, the formation of new secondary structures oriented along the friction direction, different from the initial polymer matrix, was revealed. The presence of wear products (oxidized polymer groups) and $CaSiO_3$ on the friction surfaces was recorded by infrared spectroscopy. It was established that the synthesized $CaSiO_3$ particles were deformed under the action of shear forces and participated in tribochemical processes.

**Keywords:** ultra-high molecular weight polyethylene; wollastonite; polymer composite material; wear resistance





## 1. Introduction

The ever-increasing progress in the development of modern technology requires more demanding synthesis tasks when forming special structural strength and functional properties of new materials [1]. At present and in the near future, a large share of the market for these materials will clearly be allocated to polymeric materials, in particular, composite materials. Therefore, the problem of targeted improvement of structural, functional, and operational properties of polymer composite materials (PCM) remains the most urgent task of modern material science today. In particular, to manufacture cost-effective products from PCMs, it is necessary to search for and select effective fillers that are based on both natural and synthetic raw materials. Moreover, it requires the development of new technologies that facilitate the directional variation of the properties of PCMs in a wide range of possible parameters, which can be implemented for successful and economically feasible solutions to different engineering and technical problems.

Among the most promising and widely used polymer matrices for the manufacture of PCMs is ultra-high molecular weight polyethylene (UHMWPE). Due to its high strength, low value of friction coefficient, resistance to aggressive environments, chemical inertness, etc., UHMWPE is a promising material for a wide range of fields of science and technology [2]. Lightweight, high-strength, and high-impact resistance give strategic importance to UHMWPE-based PCM, since it can be used as a basis for the manufacture of bulletproof cases of military equipment, in particular, to be used as a basic material of external structures and a frame for protection against cosmic radiation, as well as for the

manufacture of individual units and elements of parts that withstand heavy loads. For example, UHMWPE fiber boards are mandatorily used in the cabins of civil aircraft in the United States and Europe [3,4]. The high-tech application of UHMWPE composites is also relevant in biomedicine to create structural bioimplants and prostheses of high operational quality [5,6].

However, the wear resistance of UHMWPE mainly depends on its plastic deformation, which determines the formation rate of wear particles and the formation of melting defects [7]. To prevent the formation of wear particles and increase the strength of the composite based on UHMWPE, solid lubricating fillers (molybdenum disulfide, polytetrafluoroethylene, calcium stearate, graphite, boron nitride) are introduced [8,9]. The self-lubricating effect of friction was studied in [10,11] during the introduction of 2-mercaptobenzothiazole and borpolymer soft fillers into UHMWPE, while a decrease was recorded in the friction coefficient by 43% and 25%, respectively. It is known [12] that when the UHMWPE friction occurs in the subsurface layer, tribochemical reactions occur and orientation effects arise along the sliding direction due to polymer deformation. As a result of these processes, a friction film is formed in the contact zone, which protects the material from further abrasion (the so-called secondary structure). In this case, the introduced fillers affect the wear resistance of the material due to the hardening of the matrix and the change in the nature of these processes [13].

This paper considers the influence of synthetic wollastonite (CaSiO3) on the tribotechnical properties of UHMWPE. The main feature of the use of wollastonite in PCM is due to its reinforcing effect on the polymer matrix [14–16], which justifies its use as a UHMWPE modifier effectively improving the wear resistance. The introduction of wollastonite fibers at a concentration of not more than 20 wt% contributes to an increase in the wear resistance under abrasive wear [17]. In the previous paper [18], it has been shown that additional treatment of wollastonite fibers with silane, titanium, and combined silane-titanium modifiers promotes an increase in abrasion resistance of PCM (at a content of no more than 10 wt% wollastonite). The use of wollastonite as a reinforcing filler for UHMWPE was considered in the work of Panin S.V. [13]. It has been established that under mild friction conditions (P = 60 N and V = 0.3 m/s) UHMWPE/23 wt% wollastonite composite modified by the solution of KN-500 and UHMWPE/15 wt% wollastonite were noted for the maximum decrease in wear rate. At the same time, modification of wollastonite by silane coupling agents "X-550" and "Penta-1006" promotes an increase in wear resistance of PCM in hard friction conditions (P = 140 N and V = 0.5 m/s). However, in the above-stated works wollastonite of natural origin was used, on this basis, scientific and practical interest represent questions of creation of new materials with application of synthetic wollastonite on the basis of UHMWPE. Research on the interrelation of conditions of synthesis of wollastonite from various components with functional properties of PCM on the basis of wollastonite and UHMWPE are not less actual. Among the known methods of obtaining wollastonite (melt, solid-phase, hydrothermal), the preference is given to low-temperature synthesis methods. The authors of this paper have previously found [19] that filling the UHMWPE with needle $CaSiO_3$ leads to a decrease in the rate of wear, which proves the potential use of synthesized wollastonite particles. At the same time, focusing on the proven biocompatible properties of synthetic wollastonite [20–22], it should be noted that its uniqueness as a reinforcing component for UHMWPE makes it possible to create biomedical products for solving important problems related to personalized medicine use.

This paper investigates the effect of the addition of synthetic $CaSiO_3$ obtained by the hydrochemical method on the tribotechnical and thermomechanical properties of PCM.

## 2. Material and Methods

### 2.1. Materials

UHMWPE powder GUR-4022 (Celanese, Shanghai, China) with a molecular weight of 5 million g/mol and a density of 0.93 g/cm$^3$ was used for the synthesis of PCM. The choice of the UHMWPE GUR 4022 brand is due to its unique characteristics. According to the

certification of the Celanese company, UHMWPE of the GUR-4022 brand is characterized by very high abrasion resistance and increased impact resistance. Therefore, it is suitable as a self-lubricating additive for the manufacturing of heavy-duty fibers, sheet and film products, battery separators, and membrane filters [23].

The reinforcing component used is synthetic $CaSiO_3$ obtained via hydrochemical synthesis in a $CaCl_2$-$Na_2SiO_3$-$H_2O$ model system at a temperature of 20 °C [24]. A 20% solution of chemically pure $CaCl_2$ and $Na_2SiO_3$ was used as the starting components for the synthesis. A solution of $CaCl_2$ in 1.1–1.5-fold excess of the stoichiometric requirement was added to the pre-diluted $Na_2SiO_3$ solution with constant stirring. The resulting precipitate was washed with distilled water, dried at 85 °C, and then the precipitate was roasted at 850 °C for 1 h. The synthesis of calcium hydrogen silicate in the $CaCl_2$-$Na_2SiO_3$-$H_2O$ system without autoclave treatment is described by the following equation:

$$nCaCl_2 + mNa_2SiO_3 + kH_2O \rightarrow nCaO \cdot mSiO_2 \cdot kH_2O + 2mNaCl \tag{1}$$

where, $Na_2SiO_3$ is a silicate modulus equal to 1 ($SiO_2/Na_2O = 1$); at $m = n$.

### 2.2. Manufacturing of PCM

The UHMWPE and $CaSiO_3$ powders previously dried at 85 °C were mechanically mixed in a paddle mixer at a rotor speed of 1200 rpm. Furthermore, PCM was obtained by hot pressing in a vulcanization hydraulic press PKMV-100 (Impulse, Moscow, Russia) at 175 °C and a pressure of 10 MPa with a holding time of 20 min and subsequent cooling to 80 °C under pressure. The concentration of $CaSiO_3$ introduced into the UHMWPE varied between 0.5, 1, 2, 5, 10, and 20 wt%.

### 2.3. Study Methods

The tribological properties of the composites were investigated on a UMT-3 universal tribometer (CETR, Coachella, CA, USA). The test was carried out according to the finger disc friction scheme at a load of 160 N and a sliding speed of 0.5 m/s for 3 h. As a counterbody, tempered unalloyed carbon steel AISI 1045 with a hardness of 45–50 HRC and roughness of $R_a = 0.06$–0.08 μm, was used. Cylindrical samples, with a diameter of $10 \pm 0.05$ mm and a height of $20.0 \pm 1.0$ mm, were used for testing. The friction coefficient was determined in accordance with the Russian standard GOST 11629. The wear rate of the composites ($mm^3/Nm$) was calculated by the following Formula (2):

$$\text{Wear rate} = \frac{V_{lost}}{F_N \cdot d} = \frac{m_{lost}}{F_N \cdot d} \tag{2}$$

where $V_{lost}$ is the volume lost during sliding ($mm^3$); $m_{lost}$ is the mass lost during sliding (g); $\rho$ is the density of the composite; $F_N$ is the normal force (N); d is the sliding distance (m). The density of PCM was determined by the method of hydrostatic weighing in accordance with the Russian standard GOST 15139–69.

X-ray images of the synthesized $CaSiO_3$ were taken on an automatic diffractometer D8 ADVANCE (Bruker, Karlsruhe, Germany) with rotation of the sample in $CuK_\alpha$ radiation. X-ray phase analysis (XRPA) was performed using an Eva search program with the Powder Diffraction File TM XRPA database (Soorya N Kabekkodu, 2007). UHMWPE radiographs were taken on an ARL X'Tra X-ray powder diffractometer (XRD, ARL X'Tra, Thermo Fisher Scientific, Ecublens, Switzerland)). An X-ray tube with a copper anode ($\lambda(CuK_\alpha) = 0.154$ nm) was used as the radiation source. Scanning angles were from 3° to 60° with a scanning step of 0.05° and an accumulation time at each point of 3 s in reflection mode. The degree of PCM crystallinity was calculated using the Crystallinity program by the ratio of reflex areas corresponding to amorphous and crystalline regions.

The density of the obtained material was determined by a pycnometric method.

The supramolecular structure and friction surfaces of the samples were examined on a JSM-7800F scanning electron microscope (SEM) (Jeol, Akishima, Japan) in the secondary electron mode at an accelerating voltage of 1.0–1.5 kV.

IR spectra of wollastonite particles and UHMWPE, PCM in volume and their friction surfaces were obtained using a Fourier transform infrared spectrometer Varian 7000 FT-IR (Varian, Palo Alto, CA, USA) in the wave number range of 4000–500 cm$^{-1}$.

The specific area was measured using an analyzer of low-temperature nitrogen and argon adsorption-desorption at 77 K (Autosorb IQ, Quantachrome, Boynton Beach, FL, USA). The BET, SF, and HK models were used for calculating texture characteristics.

The topography of the friction surface was investigated on an atomic force microscope (AFM) NTegraPrima (NT-MTD, Moscow, Russia). Surface topography was obtained in a semi-contact scanning mode using an NSG 10 cantilever with a resonance frequency of 140–390 kHz and constant power of 3.1–37.6 N/m. The scanning scale was 50–50 μm. AFM images were processed using the Nova and Image Analysis programs (NT-MTD, Russia).

The linear thermal expansion coefficient of PCM was studied on a thermomechanical analyzer TMA-60 (Shimadzu, Kyoto, Japan) under uniaxial compression and at a constant heating rate of 3 °C/min in a helium atmosphere in accordance with ISO 11359–2:2021. Samples with dimensions of 10 × 10 × 2 mm were used for the studies. The diameter of the punch was 2.5 mm, the load on the punch was 0.50 N, and the tests were carried out in the temperature range from –80 to +160 °C.

## 3. Results and Discussion

### 3.1. Characteristics of Wollastonite

The synthesis product in the $CaCl_2$–$Na_2SiO_3$–$H_2O$ system after firing at 850 °C consists of $CaSiO_3$ triclinic modification (PDF-2, 01-084-0654) with the parameters of the crystalline cell: a = 7.92580; b = 7.32020; c = 7.06530; α = 90.055; β = 95.217; γ = 103.426 (Figure 1a). The density of wollastonite is 2.86 g/cm$^3$. The morphology of $CaSiO_3$, synthesized in this system, is in the form of porous particles of complex shapes from 10 to 50 μm (Figure 1b,c). The infrared spectrum of $CaSiO_3$ (Figure 1d) is characterized by the presence of a wide absorption band in the region of 1145–730 cm$^{-1}$, which corresponds to valence oscillations of Si–O–Si bonds and asymmetric and symmetric valence oscillations of Si–O terminal bonds of silicon tetrahedrons of wollastonite. The peaks detected at 682, 640, and 562 cm$^{-1}$ refer to the symmetrical and bending oscillations of the Si–O–Si bonds in the [$SiO_4$] tetrahedron and to the valence oscillations of the Ca–O bonds in the [$CaO_6$] octahedron [25,26].

The isotherm of low-temperature adsorption-desorption of nitrogen (Figure 2a) for the test sample of synthetic $CaSiO_3$ has a hysteresis loop corresponding to capillary condensation and belongs to type III, in accordance with the IUPAC classification characteristic of mesoporous materials [27]. At the same time, it should be noted that the sample isotherm has a sharp vertical initial section, which indicates the presence of micropores in the structure. In addition, the hysteresis loop is of a clear H3 type, which indicates the presence of slit-like pores, which is typical for materials with a laminated structure. The pore size distribution in the test sample of synthetic $CaSiO_3$ is shown in Figure 2b, which was calculated using the density functional theory method. Polymodal pore size distribution is observed without a pronounced predominant pore radius in the range of 2–5 nm (micropores) and 30–48 nm (mesopores). A small amount of pore radius in the range of 50 nm or more is also observed. The specific surface area of the sample is low and amounts to 1.3 m$^2$/g. The structured porous volume determines the bulk density of the material, which is important during its homogenization in the polymer volume, and determines its mechanical strength under cyclic loads when friction is in contact with surfaces. In this regard, the synthesized and described $CaSiO_3$ powder is attractive from the point of view of low porosity, which determines its high relative density and mechanical strength.

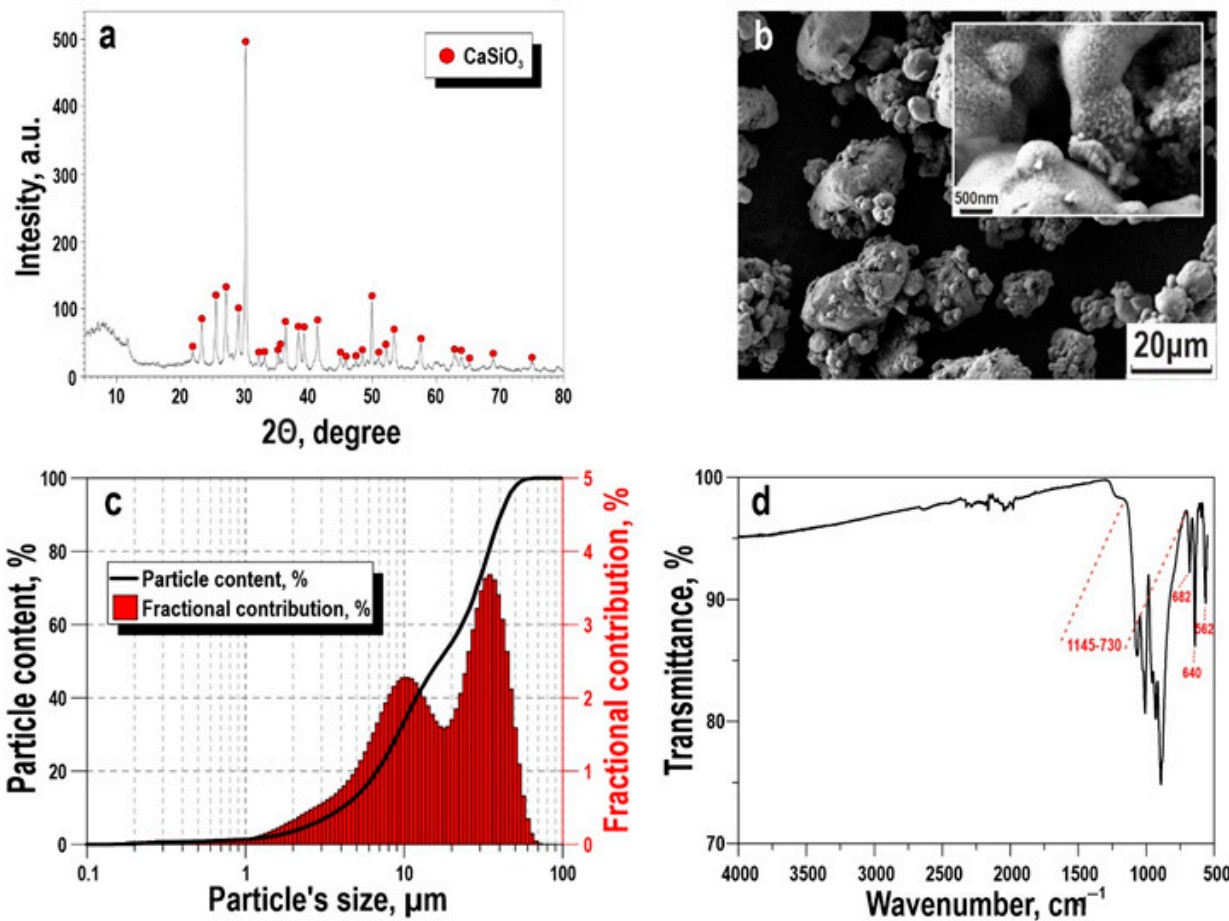

**Figure 1.** Characteristics of synthetic $CaSiO_3$ obtained in the $CaCl_2–Na_2SiO_3–H_2O$ system at 850 °C: (**a**) Phase composition of the powder composition, (**b**) SEM image of particles, (**c**) particle size distribution, (**d**) IR spectrum of the sample.

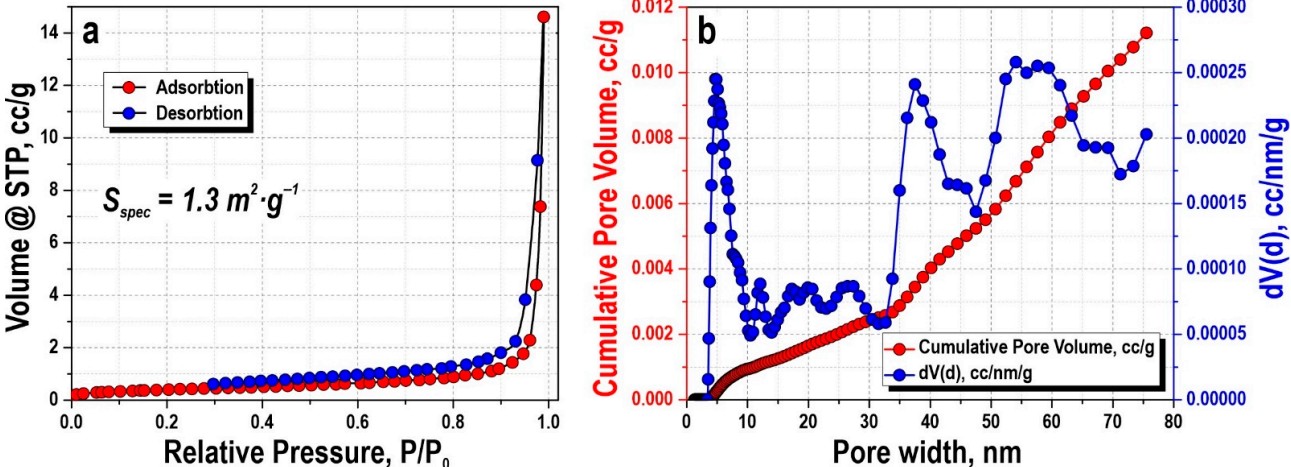

**Figure 2.** Low-temperature nitrogen adsorption-desorption curves in the presence of a sample of synthetic $CaSiO_3$ after firing at 850 °C: (**a**) isotherm and (**b**) pore size distributions.

### 3.2. Characteristics of UHMWPE

Figure 3a shows a photomicrograph of the original UHMWPE powder of the GUR 4022 grade. It is known [28] that UHMWPE grades can have different particle sizes depending on the length of polymer chains and the method of their self-assembly. Particle sizes play a key role in shaping the final properties of a material, especially in the case of filled PCM. Therefore, the UHMWPE powder used in this work is characterized by a complex hierarchical structure consisting of sub-particles that bind together through fibrillar strands "nodules" (Figure 3a) [29,30]. The powder particles have an irregular spherical shape with sizes ranging from 40 to 150 μm. Long macromolecules of UHMWPE coagulate, taking the energetically advantageous form of the "sphere", with the formation of larger agglomerates (Figure 3a). To assess the effect of the hot compression method used in the manufacture of PCM on the crystal structure of UHMWPE, an X-ray structural analysis study was carried out. It is known that UHMWPE is a semi-crystalline polymer consisting of two interpenetrating phases: Crystalline and amorphous [31,32]. The radiograph (Figure 3b) shows an amorphous halo and two intense characteristic peaks of the orthorhombic crystalline phase of UHMWPE at angles 2θ—21.5° and 24°. There was an increase in the peaks after hot pressing and a decrease in the intensity of the amorphous halo, which indicates an increase in the crystallinity degree of the initial UHMWPE composite. The degree of crystallinity of the powder was 54%, and of the composite was 58%. The IR spectrum shows (Figure 3c) that the initial UHMWPE is characterized by the presence of peaks at 2916, 2848, and 1466 cm$^{-1}$, corresponding to valence and deformation oscillations -CH$_2$ of the ethylene bond. In addition, there is a peak at 720 cm$^{-1}$ due to the pendulum oscillations of the polymer chain [33]; no other peaks were detected.

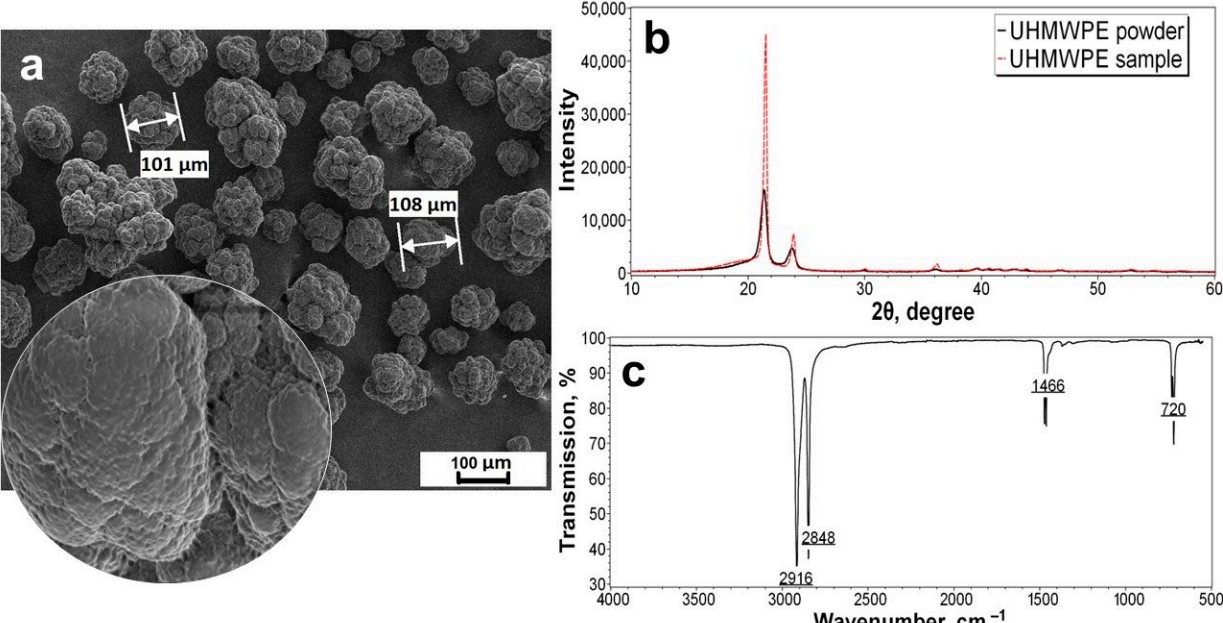

**Figure 3.** Characteristics of UHMWPE GUR 4022: (**a**) SEM images of the original powder, (**b**) radiographs of the original UHMWPE powder and sintered sample, (**c**) IR spectrum of the original UHMWPE.

Therefore, the formation of porous structures of $CaSiO_3$ and self-assembly of UHMWPE macromolecules into spherical particles were revealed.

### 3.3. Characteristics of PCM

The study results of the tribotechnical characteristics of PCM, in accordance with the content of wollastonite, are given in Table 1. An increase in the content of the $CaSiO_3$ filler leads to a decrease in the friction coefficient; the introduction of 20 wt% of $CaSiO_3$ shows a decrease of 23% compared to the original polymer. The filler in this case acts as a solid lubricant and improves the self-lubricating properties of UHMWPE, thereby facilitating shear stresses during friction [34].

**Table 1.** Tribotechnical properties and density of PCM filled with wollastonite.

| Composite | | f | L, mm | I, mg/h | K, $10^{-6}$ mm$^3$/(N·m) | ρ, g/cm$^3$ |
|---|---|---|---|---|---|---|
| Resin Torque | Wollastonite, wt% | | | | | |
| UHMWPE | 0 | $0.38 \pm 0.01$ | $0.31 \pm 0.02$ | $0.12 \pm 0.01$ | 0.57 | 0.93 |
| | 0.5 | $0.38 \pm 0.01$ | $0.21 \pm 0.02$ | $0.07 \pm 0.03$ | 0.33 | 0.94 |
| | 1 | $0.41 \pm 0.02$ | $0.12 \pm 0.01$ | $0.03 \pm 0.02$ | 0.14 | 0.94 |
| | 2 | $0.37 \pm 0.01$ | $0.09 \pm 0.01$ | $0.05 \pm 0.02$ | 0.23 | 0.95 |
| | 5 | $0.39 \pm 0.01$ | $0.13 \pm 0.01$ | $0.07 \pm 0.02$ | 0.32 | 0.96 |
| | 10 | $0.35 \pm 0.02$ | $0.10 \pm 0.02$ | $0.08 \pm 0.01$ | 0.36 | 0.99 |
| | 20 | $0.29 \pm 0.02$ | $0.15 \pm 0.01$ | $0.12 \pm 0.01$ | 0.52 | 1.03 |

Notes: f: Friction coefficient; L, mm: Linear wear; I, mg/h: Mass wear rate; K, $10^{-6}$ mm$^3$/(N·m): Wear rate; ρ, g/cm$^3$: Density of the sample.

Based on the results of the tribotechnical studies, it can be said that PCM introduced with $CaSiO_3$ has high values of wear resistance compared to the initial UHMWPE, the rate of wear in the entire concentration interval is lower than the initial UHMWPE. The maximum decrease in the wear rate (by four times) was recorded in PCM with 1 wt% $CaSiO_3$. A decrease in linear wear by a factor of 3 was recorded in PCM filled with 2 wt% wollastonite.

To explain the obtained tribotechnical results, studies of friction surfaces were carried out by methods of SEM and infrared spectroscopy. Figure 4 shows the structure of the PCM friction surfaces.

The friction surface of the PCM is characterized by the presence of grooves (Figure 4b), oriented in the friction direction, wherein there are structured secondary structures in the PCM. With an increase in the content filler, the friction surface of the PCM becomes looser, and the secondary structures become more noticeable along the friction path (Figure 4c–f). With an increase in the concentration of the filler to 2–5 wt% of the friction surface, PCM is characterized by the formation of secondary structures partially covering the surface, and the presence of wear products (frayed filler particles) was recorded (Figure 4c,d). This indicates that the $CaSiO_3$ particles deform when a load is applied and actively enter the formation of secondary structures. Therefore, the abrasive effect of the filler on the composite and on the counterbody during friction is minimized. A less rough and uniform surface is observed in the composite with 10 wt% wollastonite (Figure 4e). Solid stripes and local clusters of secondary structures are noticeable. On the surface of PCM with 20 wt% filling (Figure 4f), the presence of agglomerated $CaSiO_3$ particles reducing the intensity of shear deformations is observed.

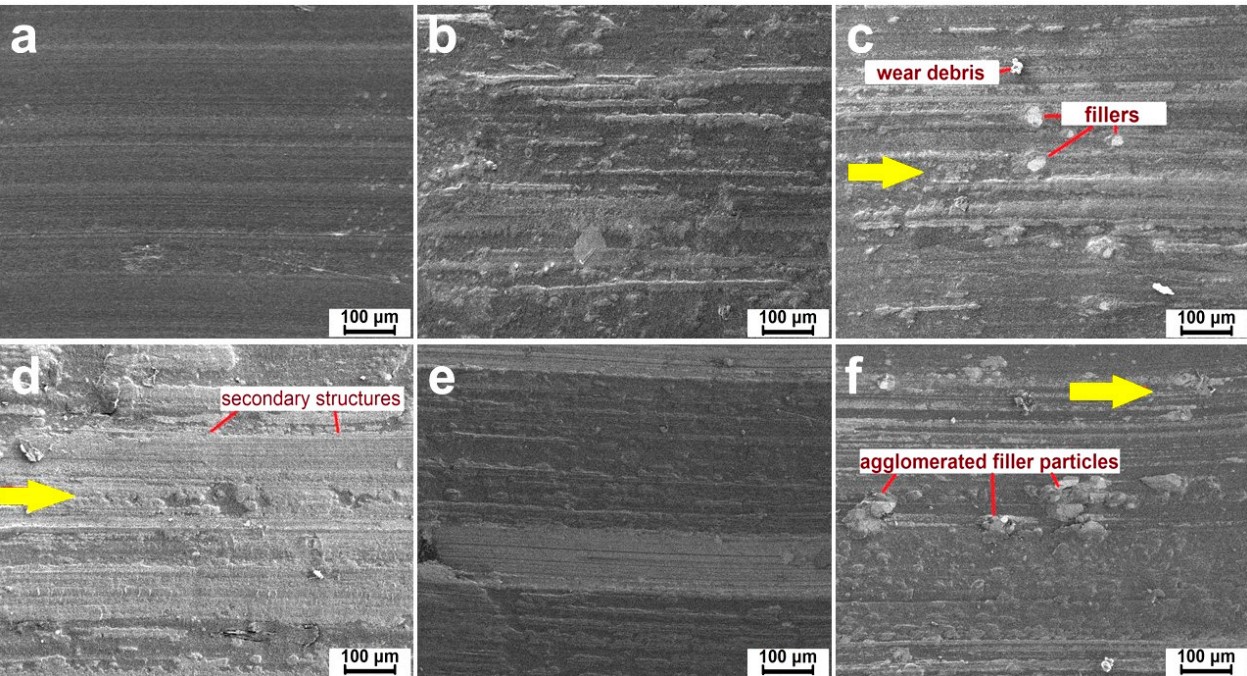

**Figure 4.** Micrographs of friction surfaces (**a**) of UHMWPE and PCM depending on the content of wollastonite: (**b**) 1 wt%, (**c**) 2 wt%, (**d**) 5 wt%, (**e**) 10 wt%, (**f**) 20 wt%.

In the infrared spectra of PCM before and after friction (Figure 5), the presence of the original UHMWPE and $CaSiO_3$ peaks was detected. The IR spectra of composites before friction (Figure 5a) show a peak at 1745 $cm^{-1}$, which corresponds to the valence of the ν(C=O) linkage of ester groups. After friction in the infrared spectra of the PCM (Figure 5b), the peak of carbonyl groups widens in the region of 1796–1500 $cm^{-1}$, which indicates the intensification of oxidative processes. In addition, the absorption band of silicon-oxygen bonds $CaSiO_3$ with an increase in the concentration of $CaSiO_3$ in PCM before and after friction in intensity does not differ. This indicates less agglomeration of wollastonite particles on friction surfaces, and a greater ability to participate in structuring processes during friction and the formation of secondary structures.

It is known [28,35] that the stability of the dimensions of the product with an increase in temperature plays a key role for the friction units, since it determines the gap between the contacting parts. Increasing the volume of PCM leads to stresses in the contact area, thereby contributing to delamination and fatigue wear. Samples from UHMWPE may be subjected to cyclic loads and various thermal stresses during friction. To assess the thermal stability of the developed materials, thermomechanical analysis (TMA) was carried out. The TMA curves (Figure 6) show that with an increase in the content of $CaSiO_3$ there is an intensive decrease in deformation with an increase in temperature to 130 °C.

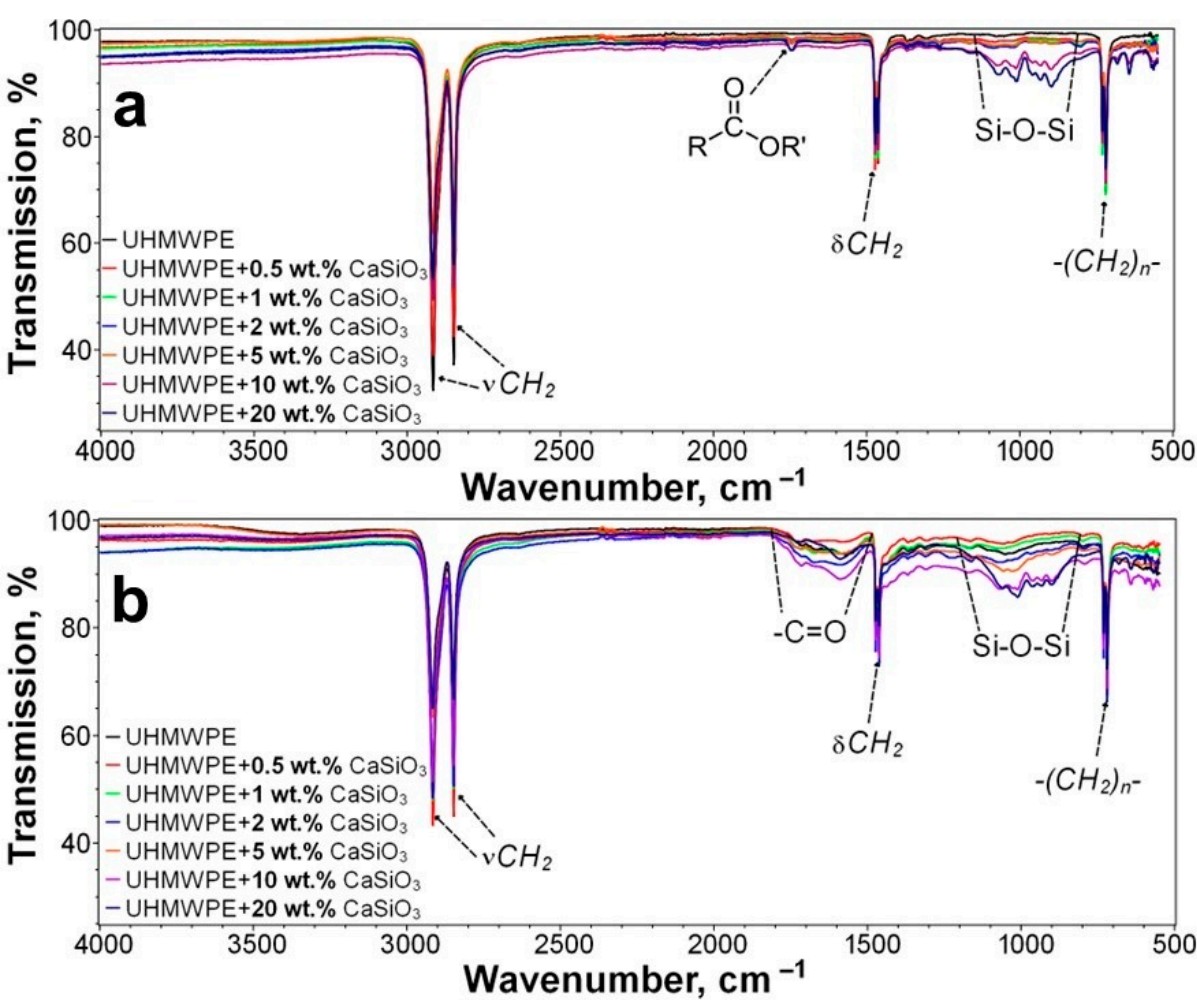

**Figure 5.** IR spectra of UHMWPE-based surfaces depending on CaSiO₃ content before (**a**) and after (**b**) friction.

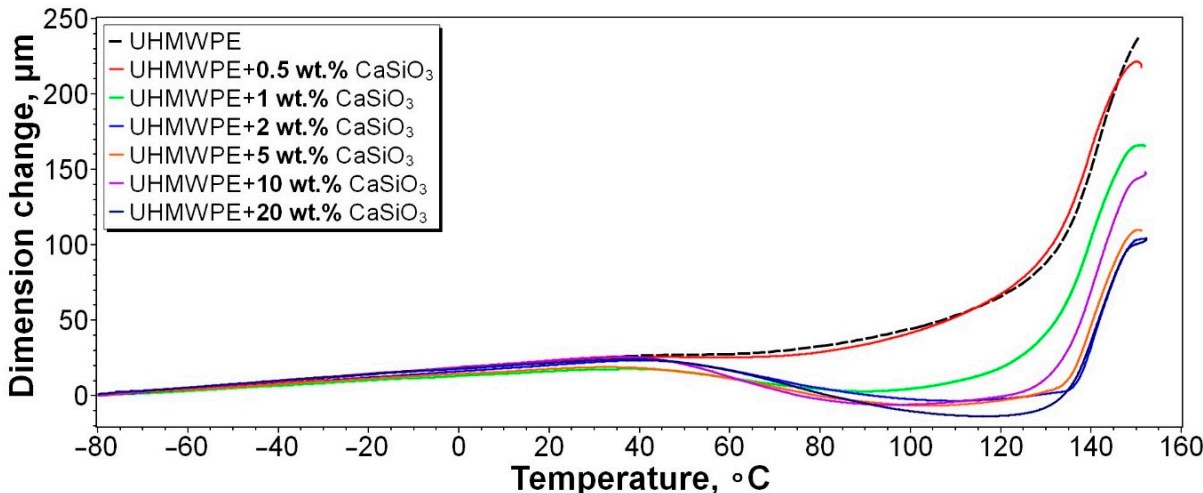

**Figure 6.** Thermomechanical curves of UHMWPE and PCM depending on CaSiO₃ content.

The introduction of 0.5 wt% $CaSiO_3$ in UHMWPE does not affect the TMA curve, which is similar to the original UHMWPE curve. When the concentration is more than 1 wt%, the PCM filler starts to decrease in volume with an increase in temperature. In this case, the melting point of PCM does not change, and the glass transition temperature shifts toward high temperatures. The decrease in the TMA curve may be due to the release of areas with zero thermal conductivity, such as air [36]. This is due to the fact that microbubbles of air and the presence of air in the pores of wollastonite may be present in the weak interfacial regions between the PCM components. It is known [35] that as a result of limiting thermal expansion in the tribocontact zone, compressive forces arise, which can lead to a plastic flow of the composite. However, in the case of composites filled with wollastonite, this process is not observed due to the high material stiffness and its compressive strength.

The study of the supramolecular structure of PCM (Figure 7) showed that $CaSiO_3$ particles contributed to the development of spherulite formations of irregular radial shape. At the same time, it can be seen that the $CaSiO_3$ particles are characterized by a weak interfacial interaction with the polymer matrix. This suggests that the friction filler particles can separate and actively participate in the formation of the transfer film on the counterbody and the secondary structures on the friction surface of the composite.

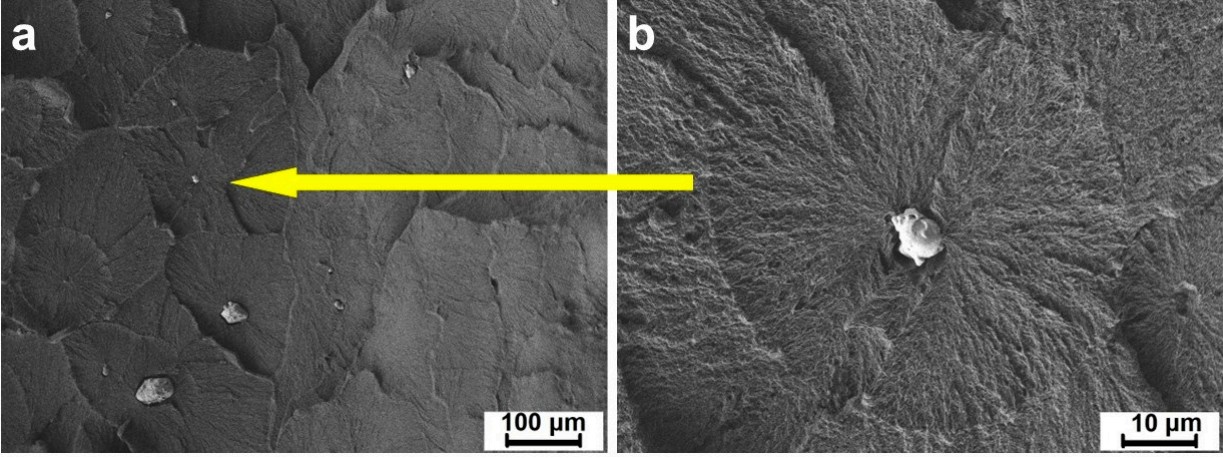

**Figure 7.** SEM images of the supramolecular structure of PCM filled with 1 wt% wollastonite, with magnification ×150 (**a**) and ×1000 (**b**).

The studies of the topography of the friction surface of the PCM (Figure 8) UHMWPE/ 20 wt% wollastonite by the AFM method revealed that there was a groove on the surface along the friction direction (friction path). In addition, the presence of secondary dense structures and filler particles with a diameter of ~5 μm, as well as small protrusions from wear products, were found. The profilogram shows that this composite has a maximum protrusion of 1.8 μm. It is known [37] that this effect of groove formation arises from the abrasive effect of the irregularities in the counterbody, the so-called microcutting process of the matrix. While the protruding regions are formed by adhesive wear, i.e., a portion of the material adheres to the surface of the steel counterbody and is cohesively cleaved off, a thin "transfer film" is formed on the metal surface as a result of this process. Furthermore, worn particles accumulate on the friction surface and, due to the participation of elements of the metal-polymer tribocoupling (filler particles, polymer matrix, and steel counterbody), form a more wear-resistant oriented intermediate film, the so-called secondary structure, a third body or tribofilm. The paper [38] reports that the thickness of the intermediate film is on average 4–9 μm. However, the thickness of this layer strongly depends on the triboloading mode and the sliding speed during friction.

Based on the studies, the following conclusion can be made regarding the patterns of friction and wear of PCM with a weak interfacial interaction at the boundary regions

(Figure 8). At the initial stages of wear and tear, the filler begins to migrate to the friction surface and peel off, causing microstresses. At the same time, the polymer matrix is deformed on the protruding microroughness of the steel counterbody, forming grooves (Figure 4). As a result of the cyclic abrasive action of the counterbody, the temperature in the contact zone increases, which leads to the formation of a transfer film on the counterbody and a secondary structure on the surface of the sample. With further PCM wear, the secondary structure is enriched with crushed and deformed $CaSiO_3$ particles. Therefore, the filler remains on the friction surface (Figure 8), and recesses are observed in the places of the detachment of the filler.

As a result, the conditions of $CaSiO_3$ synthesis and its concentration in PCM affect the tribotechnical properties and formation of the PCM friction surface structure. Therefore, in the previous work by the authors [19], when studying the influence of wollastonite synthesized in the autoclave, the active participation of wollastonite particles in friction processes was recorded. At lower concentrations, the maximum increase in wear resistance was observed, which indicated the reinforcement of the polymer matrix. However, with an increase in the critical concentration, the participation of $CaSiO_3$ in friction as abrasive particles was observed, which led to a slight decrease in wear resistance.

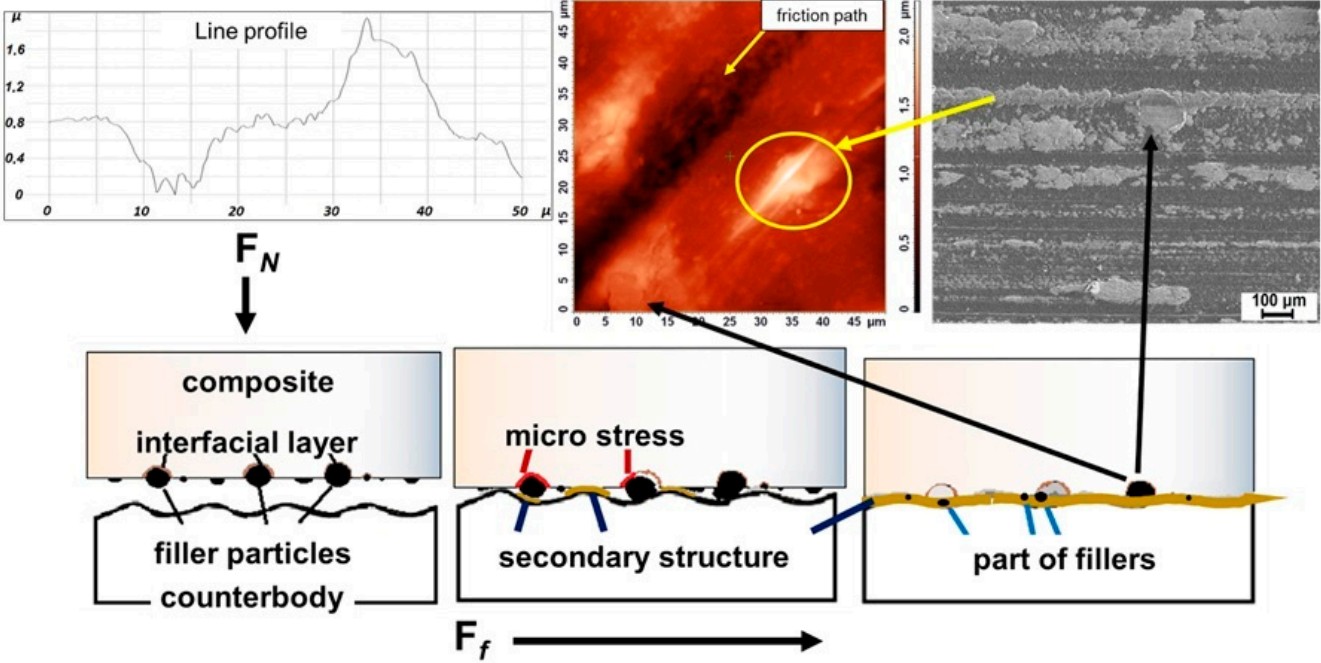

**Figure 8.** A schematic illustration of the wear of composites with weak interfacial interaction between components and the topography of the PCM friction surface filled with 20 wt% $CaSiO_3$.

## 4. Discussion

Wollastonite is one of the main micro reinforcing fillers and is widely used in various industries to modify the polymer matrix in structural materials [14,39–49]. The application of wollastonite is primarily aimed at improving the initial properties of the matrix: Mechanical, wear resistance, and thermal stability. The main studies on the influence of wollastonite on the properties of UHMWPE were carried out in a series of works by Panin S.V. [13] and Tong J. [17,18], as already noted in the introduction. When comparing the tribotechnical characteristics it is necessary to take into account the research conditions and the molecular weight of UHMWPE, as well as the nature of wollastonite and its modification. In works of Panin S.V., the molecular weight of UHMWPE was 4.5 million g/mol and tribotechnical tests were carried out in soft and hard conditions by the scheme of a friction "a shaft-cradle". The closest on conditions was hard conditions of a friction (load 140 N and speed of friction 0.5 m/s), at these conditions the best results have shown SVMPE

with 23 wt% the maintenance of wollastonite, modified "Penta-1006" (polyorganosiloxane) and "OTS" (trimethoxy(octadecyl)silane), due to strengthening of adhesion, thus untreated wollastonite does not lead to an essential increase in the wear resistance of PCM. In the works of Tong J., the wear resistance of composites was investigated at a load of 160 N and a rotational speed of 0.53 m/s, under the "shaft-pad" friction scheme. However, the molecular weight of UHMWPE was two times less than the UHMWPE GUR 4022. Proceeding from it, a comparison of tribotechnical characteristics of composites is conditional enough, and does not show in full the exact analogy with the developed composites. When investigating the effect of wollastonite on the UHMWPE matrix, the works of Yabutsuka T. should be mentioned [50], in which wollastonite was electrophoretically deposited into UHMWPE pores with further production of hydroxyapatite nucleus. Therefore, the degree of deposition and adhesion on the UHMWPE surface of wollastonite particles was investigated, while the mechanical and tribological parameters of the composites were not considered. In addition, in the previous works of other authors, a natural mineral, wollastonite, was used. In the present work, synthetic wollastonite obtained by the hydro-chemical method is used as a filler of PCM based on UHMWPE. The advantage of synthetic wollastonite is its adjustable and stable composition. In addition, the size and morphology of wollastonite particles can be varied depending on the synthesis conditions, the ratio of initial reagents, the choice of reaction medium, the presence of micro-impurities, etc. [51–53].

In [19], the effect of wollastonite obtained by the hydro-thermal method in the multicomponent system $CaSO_4-2H_2O-SiO_2-nH_2O-KOH-H_2O$ on the UHMWPE matrix was considered. The introduction of these particles leads to a decrease in the wear rate by six times relative to the original UHMWPE. Although the conditions of tribotechnical studies were the same, in this case, the synthesized wollastonite particles differ in morphology (nanosized fibers are obtained) and granulometric composition.

Increased wear resistance of the developed composites is explained by the formation of secondary structures (Figures 4 and 8). It is known [54,55] that restructuring of the structure occurs under the influence of temperature and shear deformation, and under the influence of selective wear of less stable polymer regions and gradual enrichment of the surface by more stable thermo-stable filler particles. The processes described above result in the formation of an oriented structure and discrete surface, where the filler particles take the external load. The decrease in pressure at a frictional contact is caused by a reduction in the area of actual contact, accordingly, it raises the loading ability of a material and reduces the further wear of PCM. It is known that in the process of PCM wear there are tribochemical reactions with the formation of carboxyl and hydroxyl functional groups identified by infrared spectroscopy (Figure 5) [56,57]. The oxidation mechanism of polyolefins can be characterized using the Bolland cycle (Figure 9) [58]. The scheme of UHMWPE oxidation according to the presented cycle was used in the work [59].

First, a peroxide macroradical (ROO·) is formed in the presence of air oxygen, which is easily converted to hydroperoxide (ROOH) with hydrogen detachment (inhibitory step). This process is cyclic, with the newly formed radical re-entering the reaction cycle (propagation reaction). In friction, this cycle is initiated by severe conditions: Temperature increase in the contact zone, the presence of oxygen in the air in dry friction, and the compressive force of the load [60].

In addition to the formation of hydroxy-, oxo-, and carboxy groups, the reactions of salt formation of active macroradicals with carboxy groups with counterbody metals can occur on the friction surface [61–63]. Due to the energy dispersive spectroscopic (EDS) analysis (Figure 10), the presence of the Fe element in the wear products on the friction surface of the initial UHMWPE was revealed. It confirms the participation of counterbody in tribochemical reactions at friction.

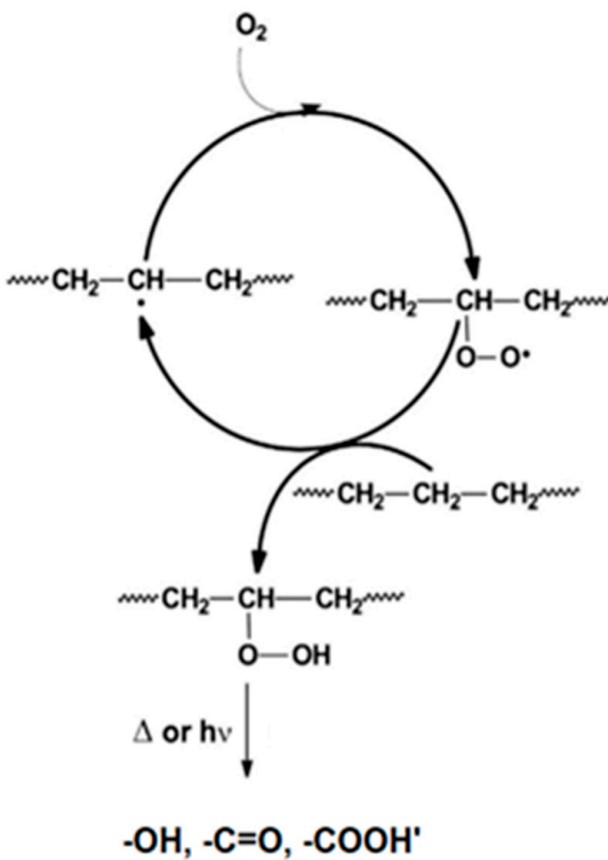

**Figure 9.** Bolland Cycles.

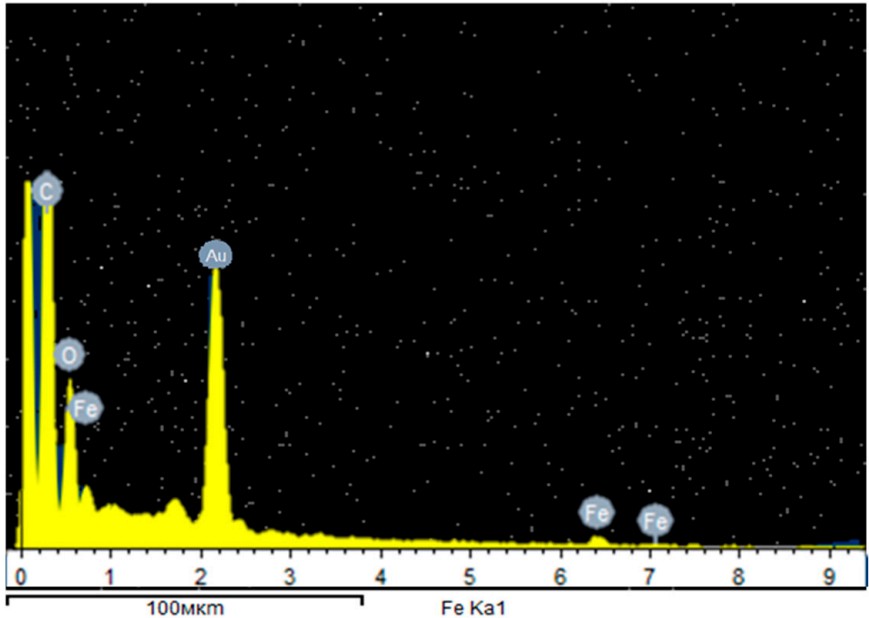

**Figure 10.** Results of elemental analysis of UHMWPE friction surface (atomic content of elements, %: C—87.95; O—9.73; Fe—2.32).

Therefore, due to the above-mentioned processes, a strong secondary structure is formed on the friction surface of PCM, which shields the material and protects it from further abrasion.

## 5. Conclusions

Based on the studies, it was established that the introduction of UHMWPE wollastonite synthesized by the low-temperature method in the solution led to a decrease in the friction coefficient by 23%, the rate of linear wear by three times, and the rate of wear by four times compared to the initial polymer. The critical concentration of wollastonite was 20 wt%, since at high concentrations, due to a weak interfacial adhesion to the matrix, the wollastonite particles peel off under the action of frictional forces and participate in the formation of the transfer film, which leads to easier relaxation and sliding processes. Therefore, this composite has a maximum reduction in the friction coefficient, but the rate of mass wear increases. The greatest increase in the relative wear resistance of PCM is recorded in composites containing 1 and 2 wt% of wollastonite. The TMA method established that composites filled with more than 1 wt% of wollastonite decreased in volume in the temperature range from −40 to +130 °C, while in the negative zone, the composites showed some elasticity. Structural studies of friction surfaces by IR spectroscopy recorded the presence of peaks in the initial UHMWPE, wollastonite particles, and new peaks of oxygen-containing groups (oxo-, carboxy-, hydroxy-), which proves the participation of the elements of the tribosystem in the course of tribochemical processes. Micrographs and the topography of PCM friction surfaces show the formation of a homogeneous structure, distinguished by the presence of grooves oriented in the direction of sliding, and with the presence of frayed wollastonite particles on the surface. In addition, the restructuring of the structure of the PCM under the influence of shear deformations and the formation of a secondary more wear-resistant surface were recorded.

**Author Contributions:** Conceptualization, S.N.D.; methodology, S.N.D., S.B.Y. and A.A.O.; software, I.Y.B. and N.N.L.; validation, I.G.Z. and E.K.P.; formal analysis, S.N.D., I.Y.B., P.S.G. and S.B.Y.; investigation, O.O.S. and S.N.D.; resources, S.B.Y. and N.N.L.; data curation, A.A.O. and S.B.Y.; writing—original draft preparation, S.N.D.; writing—review & editing, S.B.Y., O.O.S. and A.A.O.; visualization, I.Y.B. and S.N.D.; supervision, N.N.L. and P.S.G.; project administration, A.A.O.; funding acquisition, A.A.O., E.K.P. and S.B.Y. All authors have read and agreed to the published version of the manuscript.

**Funding:** The functional properties of the polymer composite material were studied with the financial support of the Ministry of Education and Science of the Russian Federation No. FSRG-2021-0016. Synthesis and characteristics of the filler were carried out within the framework of the state assignment of Institute of Chemistry, Far East Branch, Russian Academy of Sciences, project FWFN (0205)-2022-0002. Microscopic and structural studies of the polymer-composite material were carried out with the financial support of the FEFU (Priority-2030 Strategic Academic Leadership Program: Physics and Materials Science).

**Institutional Review Board Statement:** Not applicable.

**Informed Consent Statement:** Not applicable.

**Data Availability Statement:** The data presented in this study are available on request from the corresponding author.

**Conflicts of Interest:** The authors declare no conflict of interest. The funding bodies had no role in the design of the study, the collection, analysis, or interpretation of data, in the writing of the manuscript, or in the decision to publish the results.

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
