# Peer review of "A Study of the Wear Mechanism of Composites Modified with Silicate Filler"

_ceramics, doi:10.3390/ceramics5040053_

Round 1

Reviewer 1 Report

In this work, CaSiO3 was used to modify the ultra-high molecular weight polyethylene. It was found that the introduction of wollastonite into the polymer matrix contributed to a reduction in the friction coefficient by 23% and the wear rate by 4 times. In general, the authors have characterized the structure, the infrared spectra, and the tribological properties of the fabricated samples.

Some detailed comments:

1. Why did the author choose CaSiO3 to include it in the UHMWPE material system? It is suggested that the research background and current problems of this kind of materials should be strengthened in the introduction part.

2. With respect to the Characteristics of wollastonite selection (3.1), can it be assumed from Figure 1c that the size distribution of particles does not follow conventionally regular (e.g., normal distribution)? Then, are the conclusions obtained from this experiment generalizable and reproducible?

3. The authors' analysis of the experimental results is lengthy and it is recommended that the language be appropriately optimized.

In conclusion, this article is recommended to be minor revised.

Author Response

Response to the Reviewers’ Comments

Reviewer #1.

In this work, CaSiO3 was used to modify the ultra-high molecular weight polyethylene. It was found that the introduction of wollastonite into the polymer matrix contributed to a reduction in the friction coefficient by 23% and the wear rate by 4 times. In general, the authors have characterized the structure, the infrared spectra, and the tribological properties of the fabricated samples.

  1. Why did the author choose CaSiO3 to include it in the UHMWPE material system? It is suggested that the research background and current problems of this kind of materials should be strengthened in the introduction part.

Response to comment #1: Thank you for your comment! In the introduction in the paragraph, beginning with line 68, considered the rationale for the use of CaSiO3 as a modifier of UHMW PE to improve the wear resistance of composites based on them. Based on the reviewer's comments, this paragraph was corrected.

  1. With respect to the Characteristics of wollastonite selection (3.1), can it be assumed from Figure 1c that the size distribution of particles does not follow conventionally regular (e.g., normal distribution)? Then, are the conclusions obtained from this experiment generalizable and reproducible?

Response to comment #2: Thank you for your comment! A series of three powders obtained under identical conditions was used in the study, and the reproducibility of the studied properties was confirmed.

  1. The authors' analysis of the experimental results is lengthy and it is recommended that the language be appropriately optimized.

Response to comment #3: Thank you for your comment! Note taken into consideration.

In conclusion, this article is recommended to be minor revised.

Reviewer 2 Report

TITLE

The title of the manuscript is appropriate to the research described in the manuscript.

 Abstract

The abstract of this manuscript is clear and concrete on the developed topic. The information contained in this section is correctly structured and supported by the information developed in the research that supports this manuscript.

 Introduction

The introduction is clear and adequately related to the topic that is being developed in the manuscript. But it is important to improve the number of cited references and their timeliness. It is important to improve the scientific support of the contributions presented in this manuscript.

 Materials and Methods

This section is presented in a clear manner, the materials and methods used in the research presented in the manuscript are precisely specified. The reagents used in the research, their origin, quality, purity are detailed. In addition, the equipment used is described in detail, as well as its operating parameters.

 Results

This section is not defined as marked by the Journal Guidelines, so it is suggested that they adhere to the format.

 The results presented in this section are shown in a clear and orderly manner. In addition, they support the conclusions presented in the manuscript.

 But the content of this section can be improved by including elements that allow comparing the tribological behavior of these new composites with similar ones that have been developed by other research groups. This information will allow us to observe the contribution of these new composites with respect to other similar investigations in the area.

 Discussion

This section is not individually defined in the writing as marked by the Journal's guidlines. Authors are advised to adhere to the Journal format.

 Conclusion

The information presented in the conclusions of the manuscript is clear and concise. It adequately describes the experimental information developed in the research described in the manuscript. The conclusions have a solid basis regarding the premise of the manuscript and the experimental information presented.

 References

The 57.57% percent are references from the period 2017-2022, 9% from the period 2010-2015, 33.43% are references are are lower than the year 2010. In general, 41.43% of the references it is older than 5 years.

 It is important to include more references that allow comparing the impact of the filler based on synthetic wollastonite (CaSiO3) introduced in a polymeric matrix in similar tribological studies. This may provide further support to the research described in the manuscript.

Author Response

Response to the Reviewers’ Comments

Reviewer #2.

TITLE

The title of the manuscript is appropriate to the research described in the manuscript.

Abstract

The abstract of this manuscript is clear and concrete on the developed topic. The information contained in this section is correctly structured and supported by the information developed in the research that supports this manuscript.

Introduction

The introduction is clear and adequately related to the topic that is being developed in the manuscript. But it is important to improve the number of cited references and their timeliness. It is important to improve the scientific support of the contributions presented in this manuscript.

In agreement with the first reviewer's comment, we tweaked the introduction and added citations.

Materials and Methods

This section is presented in a clear manner, the materials and methods used in the research presented in the manuscript are precisely specified. The reagents used in the research, their origin, quality, purity are detailed. In addition, the equipment used is described in detail, as well as its operating parameters.

Results

This section is not defined as marked by the Journal Guidelines, so it is suggested that they adhere to the format.

 The results presented in this section are shown in a clear and orderly manner. In addition, they support the conclusions presented in the manuscript.

 But the content of this section can be improved by including elements that allow comparing the tribological behavior of these new composites with similar ones that have been developed by other research groups. This information will allow us to observe the contribution of these new composites with respect to other similar investigations in the area.

Note taken into account. A comparison with other works has been added to the "Discussions" section.

Discussion

This section is not individually defined in the writing as marked by the Journal's guidlines. Authors are advised to adhere to the Journal format.

Discussion section added.

Conclusion

The information presented in the conclusions of the manuscript is clear and concise. It adequately describes the experimental information developed in the research described in the manuscript. The conclusions have a solid basis regarding the premise of the manuscript and the experimental information presented.

References

The 57.57% percent are references from the period 2017-2022, 9% from the period 2010-2015, 33.43% are references are are lower than the year 2010. In general, 41.43% of the references it is older than 5 years.

 It is important to include more references that allow comparing the impact of the filler based on synthetic wollastonite (CaSiO3) introduced in a polymeric matrix in similar tribological studies. This may provide further support to the research described in the manuscript.

The citations are tweaked and updated.